# Characterization of Endolysin LyJH307 with Antimicrobial Activity against *Streptococcus bovis*

**DOI:** 10.3390/ani10060963

**Published:** 2020-06-01

**Authors:** Hanbeen Kim, Hyo Gun Lee, Inhyuk Kwon, Jakyeom Seo

**Affiliations:** 1Department of Animal Science, Life and Industry Convergence Research Institute, Pusan National University, Miryang 50463, Korea; khb3850@pusan.ac.kr (H.K.); ggabulzima@naver.com (H.G.L.); 2EASY BIO, Inc., Seoul 06253, Korea; eric.kwon@pathway-intermediates.com

**Keywords:** endolysin, *Streptococcus bovis*, antimicrobial agent, acute acidosis

## Abstract

**Simple Summary:**

Development of endolysin is a promising strategy because of having the ability to control problematic bacteria specifically. In this study, we developed and characterized the endolysin having lytic activity against *Streptococcus bovis* (*S. bovis*), which is one of the initiators of ruminal acidosis. Based on our findings, endolysin LyJH307 showed potent lytic activity in ruminal pH range and ruminal temperature. In addition, LyJH307 was effective against not only *S. bovis* isolated from rumen, but also several *S. bovis* groups. We suggest that LyJH307 may have a lytic effect in the ruminal condition and prevent acute ruminal acidosis by controlling *S. bovis* specifically.

**Abstract:**

*Streptococcus bovis* (*S. bovis*) is one of the critical initiators of acute acidosis in ruminants. Therefore, we aimed to develop and characterize the endolysin LyJH307, which can lyse ruminal *S. bovis*. We tested the bactericidal activity of recombinant LyJH307 against *S. bovis* JB1 under a range of pH, temperature, NaCl, and metal ion concentrations. *In silico* analyses showed that LyJH307 has a modular design with a distinct, enzymatically active domain of the NLPC/P60 superfamily at the N-terminal and a cell wall binding domain of the Zoocin A target recognition domain (Zoocin A_TRD) superfamily at the C-terminal. The lytic activity of LyJH307 against *S. bovis* JB1 was the highest at pH 5.5, and relatively higher under acidic, than under alkaline conditions. LyJH307 activity was also the highest at 39 °C, but was maintained between 25°C and 55°C. LyJH307 bactericidal action was retained under 0-500 mM NaCl. While the activity of LyJH307 significantly decreased on treatment with ethylenediaminetetraacetic acid (EDTA), it was only restored with supplementation of 10 mM Ca^2+^. Analyses of antimicrobial spectra showed that LyJH307 lysed Lancefield groups D (*S. bovis* group and *Enterococcus faecalis*) and H (*S. sanguinis*) bacteria. Thus, LyJH307 might help to prevent acute ruminal acidosis.

## 1. Introduction

Ruminal acidosis is one of the most common disorders found in the ruminant farm, and it is induced by a severe decrease of ruminal pH above the buffering capacity of ruminants when ruminants are fed high-grain diet for meeting high requirements for energy, especially early and mid-lactation periods of dairy cattle and finishing period of beef cattle. It is important that ruminal acidosis can result in a decrease in the production performance, and induce secondary metabolic diseases such as laminitis, ruminitis, and liver abscessation [1]. Although there are many standards for dividing ruminal acidosis, in general, ruminal acidosis is distinguished as sub-acute rumen acidosis (SARA) and acute acidosis based on ruminal pH range and clinical signs [2]. The reduction of ruminal pH in SARA is mainly due to the accumulation of volatile fatty acids (VFAs) by excessing the absorption capacity of rumen [3], while the onset of acute acidosis is induced by the accumulation of lactic acid in combination with a decrease of lactic acid-utilizing bacteria (such as *Megasphaera elsdenii* and *Selenomonas ruminantium*) and an increase of lactic acid-producing bacteria, especially *Streptococcus bovis* (*S. bovis*) [1,4].

The group D *S. bovis* is a gram-positive, facultative anaerobic bacterium that mainly resides in the gastrointestinal tract of humans and the rumen of ruminants. In general, *S. bovis* primarily produces acetate, formate and ethanol as main fermentation products when starch is limited or ruminal pH is higher than 6.0, whereas main fermentation product of *S. bovis* is shifted to lactate when the feeding level of starch excessively increased [5]. In the latter case, though production of ATP per glucose on *S. bovis* decreased, *S. bovis* can generate more ATP per hour, thereby inducing overgrowth of *S. bovis* [6]. For this reason, *S. bovis* has been considered as one of the initiators in acute ruminal acidosis [4,7,8].

Traditionally strict cattle management, which is labor-intensive (for example, adaptation to increasing concentrate feeds and the use of bases or buffers), has been applied to cattle to prevent ruminal acidosis [9,10]. Ionophore antibiotics such as monensin, lasalocid, and salinomycin have also been used to inhibit various Gram-positive bacteria in the rumen [11,12,13]. However, the emergence of multi-drug resistant bacteria and the risk of antibiotic residues in animal products have curtailed the use of antibiotics in livestock [14].

Endolysins, also termed phage lysins, are bacteriophage-encoded peptidoglycan (PG) hydrolases that can degrade the host bacterial PG layer at the end of the bacteriophage replication cycle [15]. Externally supplemented endolysins can attach to the PG and lyse gram-positive host bacteria that lack an outer membrane, such as a lipopolysaccharide layer in their cell walls [16]. Endolysins are promising candidates to replace antibiotics and modulate specific bacteria, owing to their high specificity, selective bacterial toxicity, rapid lytic action, and, most importantly, a low likelihood of inducing resistance. Endolysins have been developed to treat several pathologies caused by streptococci, such as meningitis (by *S. suis*), streptococcal toxic shock-like syndrome (by *S. pyogenes*), and pneumonia (by *S. pneumoniae*) [17,18,19]. However, an endolysin to control *S. bovis* remains unknown.

In recent, several endolysins developed using prophage annotation technologies have been reported [20,21]. Therefore, we expected that a novel endolysin against *S. bovis* can be developed using genomic information of the *S. bovis* prophage.

Thus, the objective of this study was to develop a novel endolysin (LyJH307) having specific lytic activity against *S. bovis* using annotation techniques and evaluate the optimal lytic condition and lytic spectrum of LyJH307.

## 2. Materials and Methods 

### 2.1. Bacterial Strains and Growth Conditions

A recombinant endolysin to inhibit *S. bovis* (LyJH307) was cloned and expressed in *Escherichia coli* (*E. coli*) DH5α and *E. coli* BL21 (DE3), respectively. *S. bovis* JB1 (ATCC^®^ 700410™), *S. equinus* (ATCC^®^ 15351™), and *S. infantarius* subspecies *infantarius* (ATCC^®^ BAA-102™) were obtained from the American Type Culture Collection (ATCC; Manassas, VA, USA) and *Bacillus subtilis* (*B. subtilis*, KCTC 3014), *S. sanguinis* (KCTC 3284), *S. mutans* (KCTC 3065), *S. alactolyticus* (KCTC 3644), *S. gallolyticus* subspecies *pasteurianus* (KCTC 3878), and *Enterococcus faecalis* (*E. faecalis*, KCTC 5191) were obtained from the Korean Collection of Type Cultures (KCTC; Jeongeup-si, Korea). All five strains isolated from Korean native cattle (Hanwoo) and goat were indicated as *S. equinus* CG14, *S. lutetiensis* HCD23-3, *S. equinus* DMF7, *S. equinus* HCD42-2, and *S. lutetiensis* HCD23-1 based on the 16s rRNA sequence similarity results using Basic Local Alignment Search Tool of National Center for Biotechnology Information (NCBI). *S. bovis* JB1 served as an indicator of LyJH307 lytic activity and was grown anaerobically in minimal medium containing (L^−1^): 292 mg of K_2_HPO_4_, 292 mg of KH_2_PO_4_, 480 mg of (NH_4_)_2_SO_4_, 480 mg of NaCl, 100 mg of MgSO_4_·7H_2_O, 64 mg of CaCl_2_·2H_2_O, 1 g of Hemin, 10 mL of Wolfe’s vitamin solution, 4 g of glucose, and 0.6 g of L-cysteine·HCl (pH 6.7). All other streptococcal species were grown in brain heart infusion (BHI) broth (Difco Laboratories Inc., Detroit, MI, USA) under anaerobic conditions and *B. subtilis* and *E. faecalis* were grown aerobically in BHI broth. Anaerobic buffer was dispensed under O_2_-free CO_2_ into glass tubes (18 × 150 mm) that were sealed with butyl rubber stoppers.

### 2.2. Identification, Cloning, and Overexpression of Recombinant LyJH307

LyJH307 was isolated from the whole genome sequence of *S. bovis* strain MPR4 deposited in GenBank (Accession Number: NZ_FOOP00000000.1) using a Rapid Annotations using Subsystems Technology (RAST) server [22]. The chemically synthesized LyJH307 gene was amplified by the polymerase chain reaction (PCR) using HiPi™ plus thermostable Taq DNA polymerase (Elpis-biotech, Daejeon, Korea) and the primers LyJH307_BamH1_F (5′-GGGGGATCCATGAATACAGATGTTTTAATCAATTGG-3′) and LyJH307_Xho1_R (5′-CCCCTCGAGTTACTTGTAATAATTGACCAAATCG-3′), which introduced a TAA stop codon. We also added BamH1 and Xho1 restriction sites to the 5ʹ and 3ʹ ends of the products, respectively. Purified DNA fragments were digested using the restriction enzymes BamH1 and Xho1 (New England Biolabs Inc., Ipswich, MA, USA), then cloned into the expression vector pET28b (Novagen Inc., Madison, WI, USA) containing an N-terminal hexahistidine-tag (6xHis tag) sequence. The cloned plasmid was transformed into competent *E. coli* BL21 (DE3) cells that were grown in Luria–Bertani medium (Difco Laboratories Inc.) until the optical density at 600 nm (OD_600nm_) reached 0.4. Thereafter, 1 mM isopropyl-β-D-thiogalactoside (IPTG) was added to the medium, and the cells were further incubated for 4 h at 37 °C. Harvested cells were suspended in lysis buffer (50 mM NaH_2_PO_4_, 300 mM NaCl, 10 mM imidazole, and pH 8.0), and lysed by sonication (KYY-80, Korea Process Technology Co., Ltd., Seoul, Korea). After centrifugation at 10,000× *g* for 15 min, the supernatant was passed through Ni-NTA Agarose (Qiagen GmbH, Hilden, Germany) and the recombinant LyJH307 purified as described by the manufacturer, was resolved by sodium dodecyl sulfate-polyacrylamide gel electrophoresis (SDS-PAGE). The purified endolysin was pooled and dialyzed against elution buffer containing (L^−1^): 50 mM NaH_2_PO_4_, 300 mM NaCl, and pH 8.0.

### 2.3. Structure and Metal Docking Site Prediction of LyJH307

The amino acid sequences of the LyJH307 were uploaded onto an Iterative Threading ASSEmbly Refinement (I-TASSER) server with standard settings to predict the secondary and three-dimensional (3D) structures of LyJH307 [23]. The predicted structures were visualized using PyMOL. The 3D modeled protein was further analyzed to find Ca^2+^ binding motifs using metal ion-binding site prediction and docking server (MIB) [24]. The MIB server was built using a fragment transformation method in which the query protein was aligned to metal binding templates that were extracted from metal-bound proteins in the Research Collaboratory for Structural Bioinformatics Protein Data bank (RCSB PDB).

### 2.4. Characterization of LyJH307

The lytic activity of LyJH307 was assayed as a decrease in OD_600nm_ [25]. To determine dose-dependent responses, *S. bovis* JB1 was cultivated to an OD_600nm_ of 0.4–0.5, then harvested and resuspended in minimal medium to adjust the OD_600nm_ to 0.8–1.0. Serially diluted endolysin (20 μL, 3.125 μg/mL to 100 μg/mL of concentration) was added to the wells of 96-well plates (SPL Life Sciences Co., Ltd., Pocheon, Korea) along with cell suspensions (180 μL) and incubated at 39 °C. The OD_600nm_ values were monitored every 10 min for 30 min using an iMark microplate reader (Bio-Rad Laboratories Inc., Hercules, CA, USA). The optimal temperature was determined by measuring the lytic activity of LyJH307 (50 μg/mL) as described above at 4 °C, 25 °C, 39 °C, 45 °C, 50 °C, and 55 °C. The optimal pH was determined by suspending *S. bovis* JB1 in 50 mM sodium acetate (pH 4.5 to 5.5), 50 mM sodium phosphate (pH 6.0 to 7.5), and 50 mM Tris-HCl (pH 8.0). The influence of NaCl concentration on the LyJH307 activity was tested by adding 0, 31.3, 62.5, 125, 250, and 500 mM NaCl at the empirically determined optimal pH buffer. The effect of divalent cations was determined as described [26]. Briefly, the endolysin (50 μg/mL) was incubated with 5 mM ethylenediaminetetraacetic acid (EDTA) at 25 °C for 30 min to chelate divalent cations attached to the endolysin. The EDTA was removed by replacing the buffer with the empirically determined buffer at the optimal pH using Amicon Ultra-4 (10 kDa) (Merck KGaA, Darmstadt, Germany) [27]. The lytic activities of the endolysin incubated with EDTA, 10 mM CaCl_2_, or 10 mM MgCl_2_ were assessed. All these experiments were conducted in triplicate.

### 2.5. The Spectrum of Lytic Activity

Bacterial cells grown as described above to an OD_600nm_ of 0.4–0.5, were harvested and resuspended in minimal medium to adjust the OD_600nm_ to 0.8–1.0. Endolysin (50 μg/mL) was then added to 96-well plates (SPL Co. Ltd., Pocheon-si, Korea) along with the cell suspension and incubated at 39 °C. The OD_600nm_ values were monitored after 1 h in the iMark microplate reader to evaluate the lytic spectrum of LyJH307.

### 2.6. Optical Microscopy

The lytic activity of LyJH307 on *S. bovis* JB1 was visually assessed using optical microscopy as follows. The *S. bovis* JB1 prepared as described above was resuspended in 50 mM sodium phosphate buffer (pH 6.0); then 45 μL of the suspension was mixed with LyJH307 treated with 10 mM CaCl_2_ (5 μL). The activity was assessed using a CKX53 phase-contrast optical microscope (Olympus, Tokyo, Japan) equipped with Tech Xcam-III (Techsan Co., Ltd., Pusan, Korea).

### 2.7. Statistical Analysis

Statistical analysis was conducted using R software (R version 3.6.3, R Foundation for Statistical Computing, Vienna, Austria). For the comparison of differences in lytic activity of LyJH307 under different characterization conditions, we used a non-parametric Kruskal–Wallis test using the kruskal.test function because residuals did not follow the normal distribution after various transformation (log, square-root, or arcsine). Differences among different groups were compared with the Dunn’s multiple comparison using dunnTest function from the FSA package, if a significant effect was observed. All *p*-values were adjusted by the Benjamini–Hochberg false discovery rate. A statistical significance was declared at *p* < 0.05. 

## 3. Results

### 3.1. Sequence Analysis and Overexpression of LyJH307

Amino acid sequencing using NCBI conserved domain database showed that LyJH307 had a modular design with two distinct domains, namely, the NLPC/P60 superfamily (cl21534, *e*-value = 1.45 × 10^−19^) with a hydrolytic function at the N-terminal, and the Zoocin A target recognition domain (Zoocin A_TRD) superfamily (cl25103, *e*-value = 1.81 × 10^−25^) at the C-terminal that might be involved in recognizing and binding the PG layer (Figure 1A).

According to the primary structure, the calculated isoelectric point of LyJH307 was 4.47, and the instability index was smaller than 40 (instability index of LyJH307 = 26.57), indicating that LyJH307 may be a stable form of the protein [28]. The secondary structure of LyJH307 determined by the I-TASSER server [23], consisted of an alpha-helix (17%), a beta-strand (20%), and a coil (63%) (Figure 1B). The HMMTOP server that predicts transmembrane helices and topology did not detect a transmembrane helix. Considering a lower B-factor value, we assumed that most residues in LyJH307 were stable (Figure 1B). Figure 1C shows the predicted 3D model of LyJH307 in a ribbon form, and Figure 1D shows a Connolly surface representation of LyJH307 in PyMOL.

Recombinant LyJH307 was expressed in *E. coli* BL21 (DE3) and purified by nickel affinity chromatography via the N-terminal 6xHis tag. The major band of purified soluble LyJH307 endolysin resolved on SDS-PAGE at a molecular mass of 32 kDa (Figure 2A).

### 3.2. Characterization of Recombinant LyJH307

We selected *S. bovis* JB1 as the reference strain for measuring the lytic activity of LyJH307 because it is highly prevalent in the rumen. LyJH307 clarified the culture media of *S. bovis* JB1 (Figure 2B) and dose-dependently reduced the optical density of the *S. bovis* JB1 above a concentration of 3.125 μg/mL (Figure 2C). Both LyJH307 levels (50 and 100 μg/mL) effectively inhibited *S. bovis* JB1; therefore, we characterized the activity of LyJH307 at 50 μg/mL. Moreover, *S. bovis* JB1 started to undergo lysis after 2 min, and most *S. bovis* JB1 cells were removed within 10 min by LyJH307 (Figure 3).

We assessed LyJH307 activity against *S. bovis* JB1 in the presence of variable pH, temperatures, salt concentrations, and metal cations to determine the optimal conditions for lysis. Lytic activity was the highest at pH 5.5 (generally high between 4.5 and 8.0) (Figure 4A) and 39 °C (maintained between 25 °C and 55 °C) (Figure 4B). Salt concentrations from 0 to 250 mM NaCl did not adversely influence LyJH307 lytic activity, but the lytic activity on the highest concentration of salt is significantly lower than 31.5 mM concentration of salt (Figure 4C). Previous studies have shown that the activity of endolysins against streptococci such as Ply700, B30, and Ly7917, is dependent on calcium cations [29,30,31]. Therefore, we investigated the effects of metal cations on the LyJH307 enzyme activity. We initially incubated LyJH307 with EDTA (5 mM) for 30 min to remove residual divalent metals. This procedure decreased LyJH307 activity by ~57%, indicating that LyJH307 requires metal ions to exert lytic activity (Figure 4D). Adding Ca^2+^ significantly restored the activity of LyJH307 that had been reduced by EDTA, by ~67% (Figure 4D). We incubated LyJH307 with Mg^2+^ to determine whether its activity is specifically Ca^2+^-dependent. Magnesium ions did not affect the activity of the LyJH307 incubated with EDTA, indicating that the activity is indeed Ca^2+^ dependent. We investigated binding and docking sites of Ca^2+^ on LyJH307 using MIB to support our findings [24]. We predicted 10 Ca^2+^ docking sites on LyJH307 and visualized them using PyMOL (Figure 5). The binding score was the highest at aspartate 132nd and glutamate 133rd in the protein sequence, reaching 1.526 (Figure 5A).

### 3.3. Antimicrobial Spectrum of LyJH307

We determined the lytic spectrum of LyJH307 against 11 Gram-positive and two Gram-negative bacteria using turbidity reduction tests (Figure 6). Among the Gram-positive bacteria, LyJH307 potently lysed *S. bovis* JB1, *S. equinus*, *S. gallolyticus* subspecies *pasteurianus*, *S. infantarius* subspecies *infantarius*, *S. sanguinis*, and all *Streptococcus* species isolated from the rumen (relative lytic activity, >90%). LyJH307 was moderately lytic against *S. alactolyticus* and *E. faecalis* (relative lytic activity range, 40–60%) and had <25%, relative lytic activity against *S. mutans* (KCTC 3065) and *B. subtilis* (KCTC 3014). However, Gram-negative bacteria were not affected by LyJH307.

## 4. Discussion

The isolation of bacteriophages with specific lytic activity against bacteria is generally a prerequisite to developing endolysins, but we aimed to create an endolysin using genomic data from the NCBI nucleotide database. We successfully identified an endolysin, LyJH307, from the whole genome sequence of te *S. bovis* strain MPR4 using the RAST server [22], and we firstly characterized a novel endolysin LyJH307 having potent lytic activity against *S. bovis* group organisms.

Endolysins taking lytic activity against Gram-positive bacteria generally have at least two functional conserved domains, an N-terminal enzymatically active domain (EAD) and a C-terminal cell-wall binding domain (CBD) [15]. Therefore, having both of them is important to predict the effective lytic activity of endolysin against target bacteria. Bioinformatic analyses using NCBI conserved revealed that LyJH307 had an N-terminal EAD comprising the NLPC/P60 superfamily and a C-terminal CBD comprising Zoocin A_TRD, thereby inferring a general type of endolysin that targets gram-positive bacteria [16]. The NLPC/P60 superfamily plays various roles in the dynamics of bacterial cell walls [32], and it hydrolyzes D-γ-glutamyl-meso-diaminopimelate or N-acetyl-muramate-L-alanine linkages within the PG stem peptides [32,33]. The NLPC/P60 superfamily generally has three significant motifs in its domains, namely, an N-terminal cysteine, a central glycine, and a C-terminal histidine that are involved in catalytic activities [32], and the EAD of LyJH307 also had three significant motifs in the catalytic domain (data not shown). The main role of CBD is to recognize and bind to ligands within the PG layer of target bacteria, thereafter, helping the EAD to act effectively [16]. The C-terminal Zoocin A_TRD superfamily is the CBD of Zoocin A, which is an exoenzyme secreted by *S. equi* subspecies *zooepidemicus* 4881 [34]. Zoocin A has been known to have lytic activity against several streptococci that are associated with a streptococcal sore throat and dental caries [34,35]. Therefore, we used turbidity reduction assays to determine LyJH307 lytic activities against several streptococcal species. The results showed that LyJH307 (25 μg/mL) killed *S. bovis* JB1 within 30 min (decrease >60%). LyJH307 activity was relatively higher under acidic than under alkaline conditions, remained high between 25 °C to 55 °C, and was dependent on the presence of Ca^2+^. A high dose of LyJH307 under optimal conditions not only clarified *S. bovis* broth but also decreased viable bacterial cells within 10 min. A high starch diet is essential to ruminants for high production but this can induce the rapid growth of amylolytic bacteria (especially *S. bovis*), reduce fibrolytic bacteria, and decrease ruminal pH when ruminants are not adapted to high-grain diets [1]. The retained bactericidal activity at acidic pH and rumen temperatures indicated that LyJH307 might be a good candidate antimicrobial molecule to prevent acute acidosis by specifically controlling *S. bovis*. 

Viridans streptococci have been classified into Mitis, Sanguinis, Mutans, Salivarius, Anginosus, and Bovis categories [36]. The *S. bovis* group comprises *S. bovis*/*equinus*, *S. infantarius* subspecies *infantarius*, *S. lutetiensis*, *S. alactolyticus*, *S. gallolyticus* subspecies *gallolyticus*, *S. gallolyticus* subspecies *macedonicus*, *S. gallolyticus* subspecies *pasteurianus* [37]. Recently, not only a pathogenetic role of *S. bovis* group in bacteremia but also the appearance of antibiotic-resistant *S. bovis* group has been emphasized [37]. Considering that LyJH307 was potently lytic against all *S. bovis* strains isolated from the rumen and commercial *S. sanguinis* and *S. bovis* groups except *S. alactolyticus* and *S. mutans*, thus, LyJH307 can be used as an effective replacement of antibiotics. In addition, LyJH307 was moderately lytic against *E. faecalis*. Zoocin A has major lytic activity against *S. pyogenes*, *S. gordonii*, and *S. mutans*. The Zoocin A consists of N-terminal M23 (EAD) having high similarity to lysostaphin, and C-terminal Zoocin A_TRD (CBD) is linked by a threonine-proline rich putative linker sequence [38]. However, the homology was only observed in the Zoocin A_TRD domain between LyJH307 and Zoocin A (32.1%, amino acid basis). Therefore, the low homology might explain the difference in the lytic spectrum. In addition, considering that the Lancefield group D (*E. faecalis* and *S. bovis* group) and H (*S. sanguinis*) organisms were lysed by LyJH307, the CBD of LyJH307 might recognize the carbohydrate moiety of bacterial antigens located on the PG of the Lancefield groups D and H [36]. Further studies are needed to verify the effect of endolysin LyJH307 against *S. bovis* and impact on rumen microbiota in the ruminal anaerobic condition.

## 5. Conclusions

In the present study, we firstly developed a novel endolysin LyJH307, using a genomic database without searching for a bacteriophage, having potent lytic activity against *S. bovis* group organisms. LyJH307 had a unique domain combination (NLPC/P60 with Zoocin A_TRD) but its lytic spectrum differed from that of the bacteriocin Zoocin A. LyJH307 was highly lytic at pH 5.0–6.5 and a ruminal temperature of 39 °C, and its activity was Ca^2+^ dependent. Therefore, LyJH307 might also be a good candidate with which to prevent acute ruminal acidosis and control bacteremia caused by antibiotic-resistant *S. bovis* strains.

## Figures and Tables

**Figure 1 animals-10-00963-f001:**
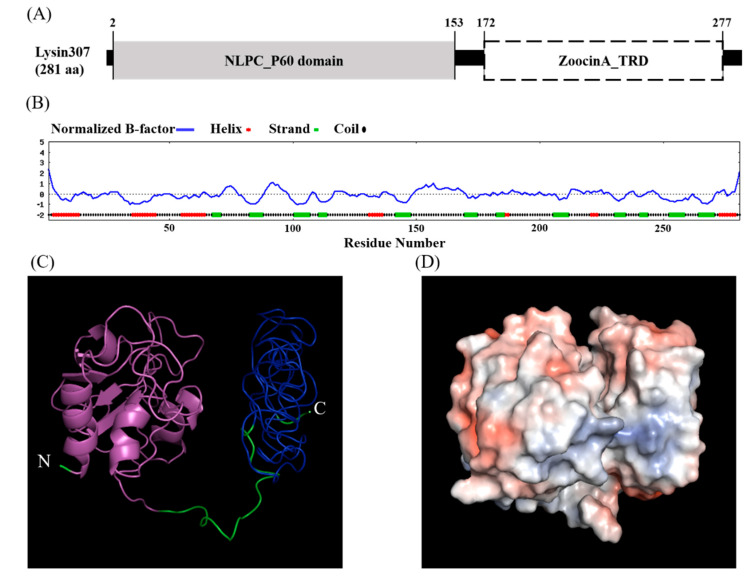
Domain and structure analysis of LyJH307. (**A**) Conserved domain of LyJH307. Gray square represents the N-terminal enzymatically active domain (NLPC/P60), and white square describes the C-terminal cell wall binding domain (Zoocin A target recognition domain, Zoocin A_TRD); (**B**) Normalized B-factor and secondary structure region of LyJH307 from Iterative Threading ASSEmbly Refinement (I-TASSER) server. (**C**) Three-dimensional model of LyJH307 made using I-TASSER server. Ribbon form shows NLPC/P60 domain (magenta) and Zoocin A_TRD (blue) of LyJH307; (**D**) Connolly surface of LyJH307 created by PyMOL (blue and red, most positive and negative polar activities, respectively).

**Figure 2 animals-10-00963-f002:**
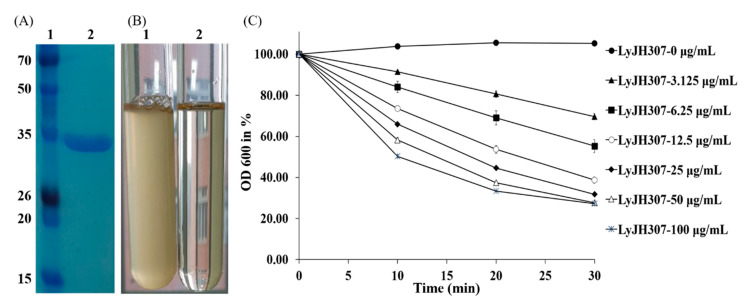
LyJH307 purification and lytic activity against *Streptococcus bovis* JB1 (ATCC^®^ 700410™). (**A**) Purified LyJH307 resolution on a 15% SDS-PAGE. Lane 1 was stained protein molecular weight markers, and Lane 2 was purified LyJH307; (**B**) Lysis of *S. bovis* JB1 by the purified LyJH307 protein; (**C**) Lytic activity of LyJH307 against *S. bovis* JB1. *S. bovis* JB1 was incubated with various doses of LyJH307 or elution buffer. Data are presented as means ± standard deviation of triplicates. OD, optical density.

**Figure 3 animals-10-00963-f003:**
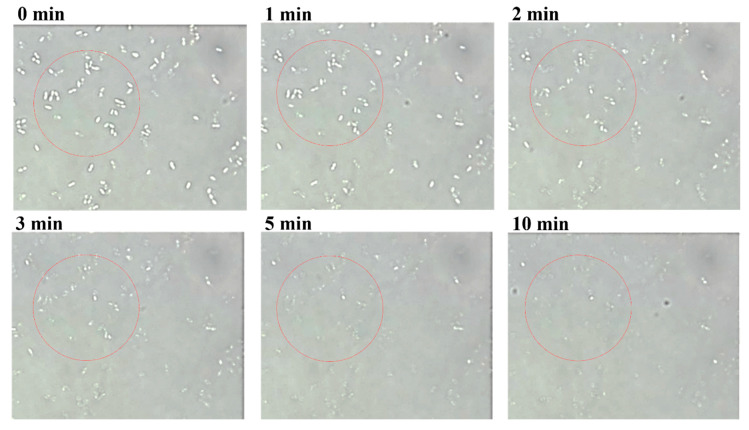
Optical microscopy of lytic activity of LyJH307 against *Streptococcus bovis* JB1 (ATCC^®^ 700410™). *S. bovis* JB1 was cultured overnight, harvested, and suspended in 50 mM sodium phosphate buffer (pH 6.0). Suspensions of *S. bovis* JB1 (45 μL) were mixed with LyJH307 after incubation with 10 mM CaCl_2_ (5 μL, 50 μg/mL) and visualized at 1000× magnification. The white one in the red ring is live cell of *S. bovis* JB1, and the disappearance of the white one in the red ring means the death of the cell.

**Figure 4 animals-10-00963-f004:**
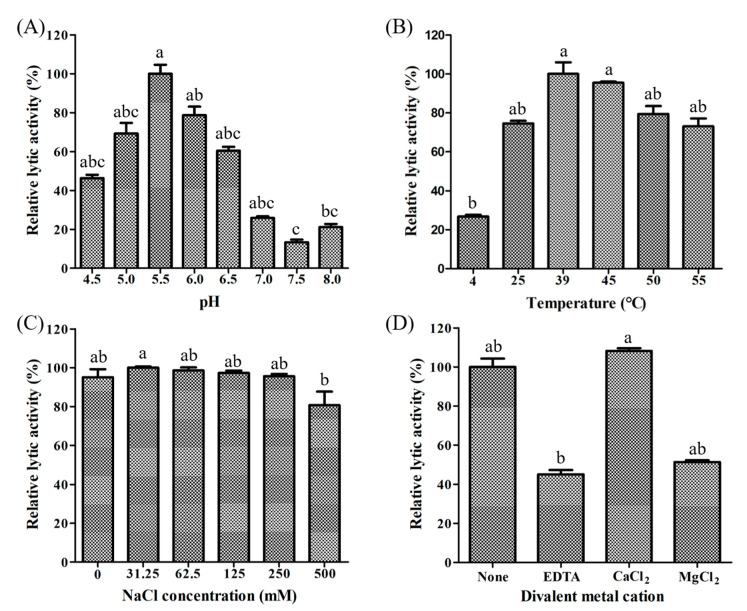
Identification of optimal conditions for the lytic activity of LyJH307 against *Streptococcus bovis* JB1 (ATCC^®^ 700410™). *S bovis* JB1 was incubated with 50 μg of LyJH307 at various (**A**) pH, (**B**) temperatures, (**C**) NaCl concentrations, and (**D**) metal ions. Data are shown as means ± standard deviation of triplicate samples.

**Figure 5 animals-10-00963-f005:**
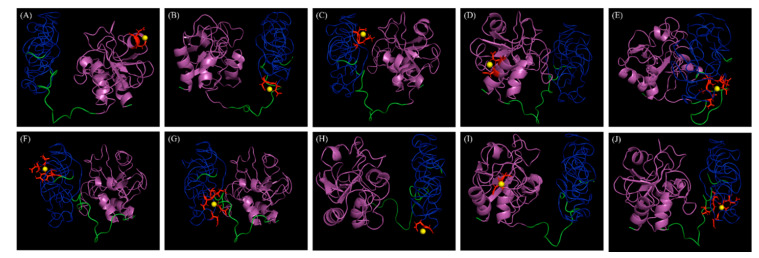
Docking positions of calcium metal ions (yellow spheres) in the three-dimensional model of LyJH307. (**A**) Binding residues, 132D and 133E; Template, Research Collaboratory for Structural Bioinformatics Protein Data Bank (RCSB PDB) ID 1ayoB0; Score, 1.526. (**B**) Binding residues, 212D and 213E; Template, RCSB PDB ID 1je5B0; Score, 1.419. (**C**) Binding residues, 272D and 273K; Template, RCSB PDB ID 1jhnA0; Score, 1.391. (**D**) Binding residues, 8N and 11E; Template, RCSB PDB ID 1e7dA0; Score, 1.338. (**E**) Binding residues, 168S, 169K, 214N, 235E, and 237D; Template, RCSB PDB ID 1k9uA0; Score, 1.329. (**F**) Binding residues, 189D, 190Y, 197D, and 199T; Template, RCSB PDB ID 1ktwA1; Score, 1.291. (**G**) Binding residues, 167D, 214N, 237D, 238E, and 261G; Template, RCSB PDB ID 1i82A0; Score, 1.290. (**H**) Binding residues, 220D and 221G; Template, RCSB PDB ID 1fi5A0; Score, 1.253. (**I**) Binding residues, 29D and 30G; Template, RCSB PDB ID 1fbl_3; Score, 1.251. (**J**) Binding residues, 168S, 214N, 235E, 237D, and 262Q; Template, RCSB PDB ID 1ht9A1; Score, 1.25. Magenta and blue represent NLPC/P60 domain and Zoocin A_TRD, respectively.

**Figure 6 animals-10-00963-f006:**
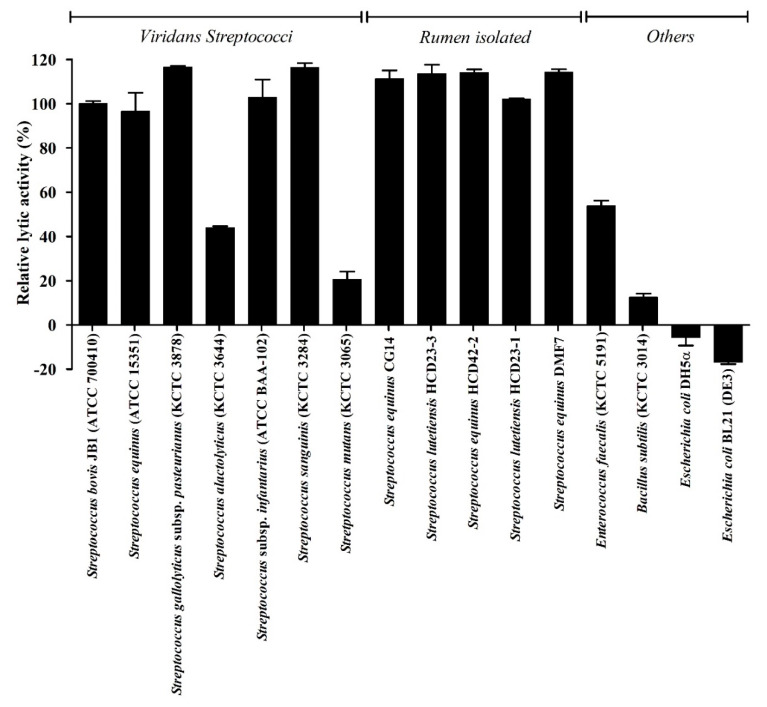
Lytic spectrum of LyJH307. Relative lytic activity (%) is defined as lytic activity against each bacterium divided by the lytic activity against *Streptococcus bovis* JB1 (ATCC^®^ 700410™). Values are shown as means ± standard deviation of triplicate samples.

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
