# Peer review of "Characterization of Endolysin LyJH307 with Antimicrobial Activity against Streptococcus bovis"

_animals, 2020, doi:10.3390/ani10060963_

Round 1
Reviewer 1 Report
This manuscript describes an investigation to the characterization of endolysin against S.bovis. This topic is interested in rumen microbes studies, is within the scope of Animal.
Detailed comments;
line 37-39 please suggest the reference regarding this sentence. Divide references(1,2) and move one of them.
line 42 updated the newly published references to contributed to a ruminal acidosis
line 53 change to more appropriate word ; critical
line 56 delete help
line 67-70 Streptococcus->S. all change to S. in content
line 117-119 Please add the information endolysin level used in this experiment
line 153 LyJH307 is stable in bacteria. What is the basis for stability?
line 181 similarly to effectively
figure 3 Can you distinguish live cell and dead in figure? please indicate using arrow mark.
line 238 add ; induce secondary
line 240 delete component of, change to an important to improve
line 268-269 a high energy diet is accomplished by adding fat diet. you should have to clarify the definition of high energy diet such as ' high starch diet.
Author Response
Response for point to Reviewer 1 Comments
Comments and Suggestions for Authors
This manuscript describes an investigation to the characterization of endolysin against S.bovis. This topic is interested in rumen microbes studies, is within the scope of Animal.
[Response] Thank you for your kind comments and suggestions. We also appreciate the time and effort you have dedicated to providing insightful feedback regarding this manuscript. Based on your suggestions, we have revised all points what you commented.
Point 1 line 37-39 please suggest the reference regarding this sentence. Divide references (1,2) and move one of them.
[Response for point 1] We have revised the sentence following your kind comment (Line 50-56).
Point 2 line 42 updated the newly published references to contributed to a ruminal acidosis
[Response for point 2] Thank you for your critical comment. We have revised the sentence and added reference (Line 56-57).
Point 3 line 53 change to more appropriate word ; critical
[Response for point 3] We have changed the word “critical” to appropriate word (Line 69).
Point 4 line 56 delete help
[Response for point 4] Sorry, we have made an error. We have removed it (Line 72).
Point 5 line 67-70 Streptococcus->S. all change to S. in content
[Response for point 5] Thank you for your kind comments. We have represented “Streptococcus” only in the first use of each species, and we have revised as abbreviated form of “Streptococcus” to “S.” in all parts.
Point 6 line 117-119 Please add the information endolysin level used in this experiment
[Response for point 6] We have added the endolysin level that we used in this experiment (Line 142).
Point 7 line 153 LyJH307 is stable in bacteria. What is the basis for stability?
[Response for point 7] Actually, the instability index means an estimation of the stability of the protein in a test tube. If the index is smaller than 40, then it means the protein may be stable. We have added not only the reference, but also some comments in the text (Line 188-189).
Point 8 line 181 similarly to effectively
[Response for point 8] We have revised as you commented (Line 219).
Point 9 figure 3 Can you distinguish live cell and dead in figure? please indicate using arrow mark.
[Response for point 9] Actually, the white one in the figure is live cell and the disappearance of white one means the death of the cell. We have added the description about the live and death cell in legend of Figure 3 (Line 224-230).
Point 10 line 238 add ; induce secondary
[Response for point 10] We have revised and moved that sentence to introduction part (Line 40).
Point 11 line 240 delete component of, change to an important to improve
[Response for point 11] Thank you for your kind comment, but we have removed that sentence because of other reviewer’s comment.
Point 12 line 268-269 a high energy diet is accomplished by adding fat diet. you should have to clarify the definition of high energy diet such as ' high starch diet.
[Response for point 12] Thank you for your critical comment. We have revised “high-energy” to “high starch” as you mentioned (Line 318).
Reviewer 2 Report
Review of Manuscript Animals-803456
The study aimed at evaluating the antimicrobial capacity of a newly developed Endolysin against Streptococcus bovis. In general, the manuscript is innovative and interesting, was well written and structured. However, the paper needs to be improved, and I would recommend the authors to consider the following remarks and I ask them to address each of my comments in their response:
Major comments
Introduction was very short and authors did not put really the study in a broad context, and authors could not really justify the realization of this study. Authors have to highlight based on literature what could be the potential in the practice of using bacteriophage in the rumen against S. bovis. I understand that there is no similar study done, but authors can make reference to those in L56-59 and give more details. Endolysins are rarely or no used in ruminant nutrition, therefore, could be helpful to give more detailed but comprehensible information about description, mode of action, production, and possible ways of application as additive of this enzymes. Finally, I would recommend authors to finish the introduction with clear objectives and expected outcomes (hypothesis)
In M&M authors described with sufficient detail and is easy to follow and I am confident that experiment and measurement were well performed. The results chapter provide precise description of the experimental results. However, the quality of some figures have to be improved and be readable for the readers. Unfortunately, I am missing statistical analyses of the assessment of the activity of the Endolysin against S. bovis under the different conditions (temperature, pH, salt concentration and metal cations). The latter is important to make sure conclusions about the Endolysin activity and must be included in the analyses of data. Indeed, authors write about some statistical differences (e.g. L197-198), but the respective p-values are not given and the statistical method used for analyses is not described in M&M.
Minor comments
L37-40: Do a better differentiation about under which conditions S. bovis produces acetate or lactate, here is confusing.
L40: What is the main substrate for S. bovis? Cereal grain is too general. Do you mean starch?
L42: Here and as well in the discussion make reference to sub-acute rumen acidosis (SARA)
L61: Here you cannot write about results you obtained
L60-61: Write clear objectives and hypothesis at the end of the introduction
L72: Specify “which other species”
L158: Improve the quality of Figure 1B
L170: Figure 2C is not readable. Improve quality
L192: I am missing here some statistics to really check whether lytic activity was affected by different pH, temperature, NaCl concentration and metal cations. This statement is only valid after the respective statistical analyses (ANOVA), the same for statement in L193-194, 197-198, 201
L237-244: Be careful of not repeating statements that were included in the introduction.
Author Response
Response for point to Reviewer 2 Comments
Comments and Suggestions for Authors
The study aimed at evaluating the antimicrobial capacity of a newly developed Endolysin against Streptococcus bovis. In general, the manuscript is innovative and interesting, was well written and structured. However, the paper needs to be improved, and I would recommend the authors to consider the following remarks and I ask them to address each of my comments in their response
[Response] Thank you for your kind comments and suggestions. We also appreciate the time and effort you have dedicated to providing insightful feedback regarding this manuscript. Based on your suggestions, we have revised all points what you commented.
Major comments
Introduction was very short and authors did not put really the study in a broad context, and authors could not really justify the realization of this study. Authors have to highlight based on literature what could be the potential in the practice of using bacteriophage in the rumen against S. bovis. I understand that there is no similar study done, but authors can make reference to those in L56-59 and give more details. Endolysins are rarely or no used in ruminant nutrition, therefore, could be helpful to give more detailed but comprehensible information about description, mode of action, production, and possible ways of application as additive of this enzymes. Finally, I would recommend authors to finish the introduction with clear objectives and expected outcomes (hypothesis)
In M&M authors described with sufficient detail and is easy to follow and I am confident that experiment and measurement were well performed. The results chapter provide precise description of the experimental results. However, the quality of some figures have to be improved and be readable for the readers. Unfortunately, I am missing statistical analyses of the assessment of the activity of the Endolysin against S. bovis under the different conditions (temperature, pH, salt concentration and metal cations). The latter is important to make sure conclusions about the Endolysin activity and must be included in the analyses of data. Indeed, authors write about some statistical differences (e.g. L197-198), but the respective p-values are not given and the statistical method used for analyses is not described in M&M.
[Response for major comments] Thank you for your critical major comments. We have improved our introduction part (Line 36-47 and 76-81) and figure quality (Figure 1 and 2) following your kind comments. We also included the statistical analysis using non-parametric statistical analysis (Kruskal-wallis test) (Line 171-179 and Figure 4).
Point 1 L37-40: Do a better differentiation about under which conditions S. bovis produces acetate or lactate, here is confusing.
[Response for point 1] We have added more detail description following your comment (Line 50-56).
Point 2 L40: What is the main substrate for S. bovis? Cereal grain is too general. Do you mean starch?
[Response for point 2] We have changed the expression to “starch” (Line 50-52).
Point 3 L42: Here and as well in the discussion make reference to sub-acute rumen acidosis (SARA)
[Response for point 3] We have added descriptions about acute acidosis and sub-acute acidosis more detail for better understanding of topic (Line 36-47).
Point 4 L61: Here you cannot write about results you obtained
[Response for point 4] Thank you for your kind comment. We have removed the sentence related to our results.
Point 5 L60-61: Write clear objectives and hypothesis at the end of the introduction
[Response for point 5] We have added the sentences containing clear objectives and hypothesis (Line 76-81).
Point 6 L72: Specify “which other species”
[Response for point 6] We have revised the sentence following the reviewer’s comment (Line 94-97).
Point 7 L158: Improve the quality of Figure 1B
[Response for point 7] We have improved the quality of Figure 1B.
Point 8 L170: Figure 2C is not readable. Improve quality
[Response for point 8] We have improved the quality of Figure 2C.
Point 9 L192: I am missing here some statistics to really check whether lytic activity was affected by different pH, temperature, NaCl concentration and metal cations. This statement is only valid after the respective statistical analyses (ANOVA), the same for statement in L193-194, 197-198, 201
[Response for point 9] Thank you for your kind comment. We have added statistical analysis using non-parametric Kruskal-wallis test because the data of characterization did not follow normal distribution even after data transformed (Log, Square-root, Arcsine). If the significance was detected, we analysed post hoc test using Dunn test (Line 171-179, Figure 3).
Point 10 L237-244: Be careful of not repeating statements that were included in the introduction.
[Response for point 10] We have removed the statements following your comment.
Round 2
Reviewer 2 Report
thanks to the authors for considering my remarks. The quality of the paper was improved and I have no more comments